

# A general pattern of the species abundance distribution

Qiang Su

College of Earth and Planetary Sciences, University of Chinese Academy of Sciences, Beijing, China

## ABSTRACT

Since the 1970s, species abundance distributions (SADs) have been one of the most fundamental issues in ecology and have frequently been investigated and reviewed. However, there was surprisingly little consensus. This study focuses on three essential questions. (1) Is there a general pattern of SAD that no community can violate it? (2) If it exists, what does it look like? (3) Why is it like this? The frequency distributions of 19,833 SADs from eight datasets (including eleven taxonomic groups from terrestrial, aquatic, and marine ecosystems) suggest that a general pattern of SAD might exist. According to two hypotheses (the finiteness of the total energy and the causality from the entropy to the diversity), this study assumes that the general pattern of SAD is approximately consistent with Zipf's law, which means that Zipf's law might be more easily to observe when one investigates any SAD. In the future, this conjecture not only needs to be tested (or supported) by more and more datasets, but also depends on how well it is explained from different angles of theories.

## INTRODUCTION

Species abundance distributions (SADs) describe the distribution of commonness and rarity in a community (*McGill et al., 2007*; *Baldridge et al., 2016a*). *McGill et al. (2007)* stated that "understanding SAD is a major stepping stone to understanding communities in general". SADs play a central role in ecology because the determinants of diversity also depend on how well SADs are explained (*May, 1975*; *Magurran, 1988*; *Tokeshi, 1993*; *McGill et al., 2007*; *Baldridge et al., 2016a*).

Since the 1970s, numerous SAD models have been proposed on various theoretical grounds, and they were observed in real situations (*May, 1975*; *Frontier, 1987*; *Magurran, 1988*; *Tokeshi, 1993*; *Mouillot et al., 2000*; *McGill et al., 2007*; *Baldridge et al., 2016a*; *Su, 2016*). However, it is difficult to draw general conclusions about which models provide the best fit to SADs (*Baldridge et al., 2016a*). Although many ecologists hoped that distinguishing subtle variations in these models would provide a decisive test, this had not worked well (*McGill et al., 2007*).

*Baldridge et al. (2016a)* pointed out that the log-series model provided a slightly better fit to the abundance distributions of over 16,000 community samples. *Connolly et al. (2014)* suggested the Poisson lognormal model as an appropriate description of 1,185 SADs from 14 marine ecosystems. A similar study according to 558 samples indicated that lognormal

Corresponding author
Qiang Su, sqiang@ucas.ac.cn

type SADs fitted much better than log-series model or the Zipf model (*Ulrich, Ollik & Ugland, 2010*). In brief, SAD models predicted very similar shapes and their distinction became problematic (*May, 1975*; *Magurran, 1988*; *Tokeshi, 1993*; *McGill et al., 2007*; *Ulrich, Ollik & Ugland, 2010*; *Connolly et al., 2014*; *Baldridge et al., 2016a*).

*Baldridge et al. (2016a)* suggested that the SAD usually did not contain sufficient information to distinguish among different models and a more promising way was to evaluate each model's ability to simultaneously explain multiple macroecological patterns. In fact, when one investigates any community, it is ubiquitously observed that many species are rare and just a few are common (*May, 1975*; *Frontier, 1987*; *Magurran, 1988*; *Tokeshi, 1993*; *Mouillot et al., 2000*; *McGill et al., 2007*; *Baldridge et al., 2016a*; *Su, 2016*). Although there is a great variability in the observed SAD, such consistent pattern appears to be very universal (*May, 1975*; *Magurran, 1988*; *Tokeshi, 1993*; *McGill et al., 2007*; *Ulrich, Ollik & Ugland, 2010*; *Connolly et al., 2014*; *Baldridge et al., 2016a*).

The purpose of this study is no longer the comparison of SAD models. The main questions are: (1) Is there (or is there not) a general pattern of SAD that no community (from the marine benthos to the Amazonian rainforest) can violate it? (2) If such pattern exists, what does it look like? (3) Why is a community generally organized in such pattern? To this end, the frequency distributions of 19,833 SADs from eight datasets (including eleven taxonomic groups from terrestrial, aquatic, and marine ecosystems) are evaluated.

## METHODS

Since SAD models all can describe the pattern that many species are rare and just a few are common, a new fractal model of SAD (*Su, 2016*) is selected for three reasons. (1) Its hypothesis is easy to fit (*Frontier, 1987*; *Mouillot et al., 2000*); (2) the frequency of SADs is easy to obtain as there is only one parameter in this model (*Su, 2016*); (3) the extension of this model into a larger ecological context and its ability to explain multiple ecological patterns have not been fully understood.

The theoretical bases of this model have been given by *Frontier (1985)*, *Frontier (1987)* and *Frontier (1994)*. In ecology, its interpretation leaded to two different but non-contradictory interpretations (*Zipf, 1949*; *Mandelbrot, 1953*; *Frontier, 1987*; *Frontier, 1994*; *Mouillot et al., 2000*; *Su, 2016*). One was the "cost of a species", which was linked with the amount of assimilated energy that it required. For example, it is costlier in terms of energy for an ecosystem to produce and maintain a carnivore than a primary producer, because of the loss of energy at each trophic level (*Frontier, 1987*; *Frontier, 1994*). The other referred to the existence of previous conditions allowing the presence of a species (*Frontier, 1987*; *Frontier, 1994*; *Su, 2016*). For example, some of these conditions may be the occurrences of some previous species in the ecological succession since any species modifies the biological and physical environment, permitting or hindering another species to appear (*Frontier, 1985*; *Frontier, 1987*; *Frontier, 1994*).

According to the original hypothesis (when $K$ more species appear at each step of the accumulation process, their abundance are $k$ times less abundant and $K = k^d$, where $d$

($>0$) is a fractal dimension (*Mouillot et al., 2000*)), SAD in a community is

$$\frac{A_r}{A_1} = r^{-p} \tag{1}$$

where $r(=1,2,3,...S)$ is the rank of species sorted down by species abundance; $A_1$ and $A_r$ are the abundance of dominant and the $r$th species; $p(=1/d)$ is the fractal parameter, which determines the pattern of the SAD (*Su, 2016*). For example, when $p=1$ and $S=6$, SAD ($A_r/A_1$) is

$$1, 1/2, 1/3, 1/4, 1/5, 1/6.$$

Let $F_r = \ln(A_r/A_1)$ and $D_r = \ln(r)$. By minimizing the sum of squared errors ($\sum_{r=1}^{S}(-pD_r - F_r)^2$), $p$ is estimated as follows

$$p = \frac{-\sum_{r=1}^{S} D_r F_r}{\sum_{r=1}^{S} D_r^2}. \tag{2}$$

Similarly, if SAD ($A_r/A_1$) in a community is 1, 1/2, 1/3, 1/4, 1/5, 1/6, the fractal $p$ is 1 according to Eq. (2).

The sum of Eq. (1) is

$$\frac{A_T}{A_1} = \sum_{r=1}^{S} r^{-p} \tag{3}$$

where $A_T$ is the total abundance.

According to Hill's notation that is related to Rényi's definition of a generalized entropy, $A_T/A_1$ is an effective number of species with the order $a = \infty$ (*Rényi, 1961*; *Hill, 1973*). When $p = 1$, the difference between $A_T/A_1$ and $\ln(S)$ in mathematic converges to the Euler–Mascheroni constant.

If $S$ is infinite, Eq. (3) is

$$\frac{A_T}{A_1} = \sum_{r=1}^{\infty} r^{-p}. \tag{4}$$

Equation (4) is consistent with the generalization of the harmonic series. It converges for all $p > 1$ and diverges for $p \le 1$. When $p = 1$, Eq. (4) is the observed Zipf's law (or Zipf distribution) (*Zipf, 1949*; *Seuront, 2009*).

In brief, Eq. (1) is based on the fractal hypothesis of diversity (*Frontier, 1987*) to create a mathematic link between the Rényi's entropy (*Rényi, 1961*; *Hill, 1973*) and an empirical distribution (Zipf's law) (*Zipf, 1949*).

## Datasets

Eight datasets (named "fish", "diatom", "nabc", "mcdb", "gentry", "fia", "cbc" and "bbs") from two sources (*Baldridge et al., 2016a*; *Passy, 2016a*) were used for four reasons. (1) These datasets are under different environments with broad representations; (2) they are relatively reliable as they have been used in SAD studies (*Baldridge et al., 2016a*; *Passy, 2016a*); (3) the frequency distributions of $p$ for these datasets are unclear; (4) published

**Table 1** The detailed information of the fractal *p* (*Su, 2016*) for eight datasets (named "fish", "diatom", "nabc", "mcdb", "gentry", "fia", "cbc" and "bbs") from two sources (*Baldridge et al., 2016a*; *Baldridge et al., 2016b*; *Passy, 2016a*; *Passy, 2016b*). The average and median value of *p* for the entire dataset are $1.108 \pm 0.003$ and 1.034, respectively. Although the range of the fractal *p* is over one order of magnitude (from 0.235 to 5.825), the average and median value of *p* for eight groups are consistent, noting that they are close to 1.

| Fractal *p* | Maximum | Minimum | Median | Average | Sample numbers |
|---|---|---|---|---|---|
| diatom | 5.825 | 0.335 | 1.272 | $1.343 \pm 0.008$ | 3,224 |
| fish | 4.563 | 0.756 | 1.592 | $1.702 \pm 0.019$ | 761 |
| bbs | 2.375 | 0.548 | 0.938 | $0.984 \pm 0.004$ | 2,769 |
| cbc | 3.738 | 0.733 | 1.492 | $1.556 \pm 0.008$ | 1,999 |
| fia | 2.229 | 0.235 | 0.907 | $0.931 \pm 0.003$ | 10,355 |
| gentry | 1.851 | 0.352 | 0.827 | $0.872 \pm 0.019$ | 222 |
| mcdb | 3.265 | 0.495 | 1.547 | $1.587 \pm 0.052$ | 103 |
| nabc | 3.112 | 0.540 | 1.240 | $1.278 \pm 0.017$ | 400 |
| Total | 5.825 | 0.235 | 1.034 | $1.108 \pm 0.003$ | 19,833 |

datasets are easy to recheck. The detail information of these datasets can be found in appendixes of *Passy (2016a)*, *Passy (2016b)*, *Baldridge et al. (2016a)* and *Baldridge et al. (2016b)*. Briefly, 19,833 quantitative samples from eleven taxonomic groups (representing over three billion individual terrestrials, aquatic, and marine organisms) were collected to explore the frequency of empirical SADs.

## RESULTS

According to Eq. (2), the fractal *p* for the entire dataset (*Baldridge et al., 2016a*; *Passy, 2016a*; *Passy, 2016b*; *Baldridge et al., 2016b*) is from 0.235 to 5.825 (Table 1). The quality of fits is measured by $R^2$ (Table 2), which denotes the goodness of fit on the log-transformed variables (log *r* and log $A_r/A_1$, please see Code 1 in Supplemental Files). The average and median value of *p* are $1.108 \pm 0.003$ and 1.034, respectively. The highest value of the mean *p* ($1.702 \pm 0.019$) is in the "fish" group (*Passy, 2016b*; *Passy, 2016a*). The lowest one is in "fia" group ($0.931 \pm 0.003$) (*Baldridge et al., 2016a*; *Baldridge et al., 2016b*). The median *p* is from 0.827 to 1.592. In short, although the range of the fractal *p* for the entire dataset is over one order of magnitude, the average and median value of *p* for eight groups are similar (close to 1) (Table 1).

The frequency distributions of the fractal *p* for eight groups are presented in Fig. 1. The "diatom" and "nabc" groups show similar frequency distributions, noting that the fractal *p* centrally occurs in the range from 1 to 1.33 (Figs. 1A and 1H); For the "bbs", "fia" and "gentry" groups, *p* is near to 1 (from 0.67 to 1, Figs. 1C, 1E and 1F); The frequency distributions of the fractal *p* for the "fish", "cbc" and "mcdb" groups skew to the higher value that *p* is from 1.33 to 1.67 (Figs. 1B, 1D and 1G). The consensus of the *p* distributions for every group is clear that *p* mostly appears close to 1. It is rare that the fractal *p* is far greater than 1 or very near 0 (Fig. 1).

Briefly, the patterns of *p* frequencies (Fig. 1) and the average and median value of *p* (Table 1) are consistent, which both suggest that *p* closer to 1 does seem to be happening

**Table 2** The goodness of fit ($R^2$, please see Code 1 in Supplemental Files) of the fractal model (*Su, 2016*) on each of 19,833 samples are shown in the following table (the range in the "Average" column is standard error). $R^2$ varies between 0 and 1, with larger numbers indicating better fits.

| $R^2$ | Maximum | Minimum | Median | Average | Sample numbers |
|---|---|---|---|---|---|
| diatom | 1.000 | 0.460 | 0.917 | 0.901 ± 0.001 | 3,224 |
| fish | 0.997 | 0.397 | 0.841 | 0.829 ± 0.004 | 761 |
| bbs | 0.982 | 0.555 | 0.792 | 0.791 ± 0.001 | 2,769 |
| cbc | 0.990 | 0.474 | 0.787 | 0.785 ± 0.002 | 1,999 |
| fia | 0.989 | 0.089 | 0.880 | 0.860 ± 0.001 | 10,355 |
| gentry | 0.969 | 0.640 | 0.911 | 0.892 ± 0.004 | 222 |
| mcdb | 0.981 | 0.466 | 0.869 | 0.838 ± 0.011 | 103 |
| nabc | 0.986 | 0.639 | 0.884 | 0.872 ± 0.004 | 400 |
| Total | 1.000 | 0.089 | 0.867 | 0.849 ± 0.001 | 19,833 |

more frequently in real situations. The mechanisms underlying the frequency distributions of $p$ might warrant further investigations.

## DISCUSSION

"We are all blind men (and women) trying to describe a monstrous elephant of ecological and evolutionary diversity" (*Nanney, 2004*; *Chao, Chiu & Jost, 2010*). No matter how diversity is defined, there is unimaginable variation in the diversity of entire living systems (*Pielou, 1975*; *Huston, 1994*). The general consensus is that an informative way to summarize the characteristics of diversity is not a statistic index (e.g., Shannon's index or Simpson's index) but the species abundance distributions (SADs) (*Pielou, 1975*; *Frontier, 1987*; *Magurran, 1988*; *Tokeshi, 1993*). Countless investigations and comparisons of the SAD models have been explored from different angles of theories (*May, 1975*; *Frontier, 1987*; *Magurran, 1988*; *Tokeshi, 1993*; *Mouillot et al., 2000*; *McGill et al., 2007*; *Baldridge et al., 2016a*; *Su, 2016*). Unfortunately, it hardly worked (*McGill et al., 2007*; *Baldridge et al., 2016a*). Thus, as noted before, the main purposes of this study shift to three fundamental questions. (1) Is there a consistent general pattern of SADs? (2) If it exists, what does it look like? (3) Why is it like this?

Firstly, according to following three reasons, this study suggests that a general pattern of SADs might exist. (1) The number of community samples in this study is relatively adequate. Nearly 20,000 quantitative samples are used to explore the frequency distributions of SADs. (2) The sources of datasets are extensive, including terrestrial, aquatic, and marine ecosystems (*Baldridge et al., 2016a*; *Passy, 2016a*; *Passy, 2016b*; *Baldridge et al., 2016b*). The above two points indicate that the datasets used in this paper have broad representation and the frequency distributions of SADs based on such datasets are reliable. (3) If the general pattern of SADs does not exist, the frequency distributions of the fractal $p$ will be discrete and irregular, or it will be quite different for eight groups. However, $p$ distributions for different taxonomic categories and groups show a consistent pattern. It is very rare that the fractal $p$ is far greater than 1 or very near 0, and $p$ closer to 1 is the most common case (Fig. 1).

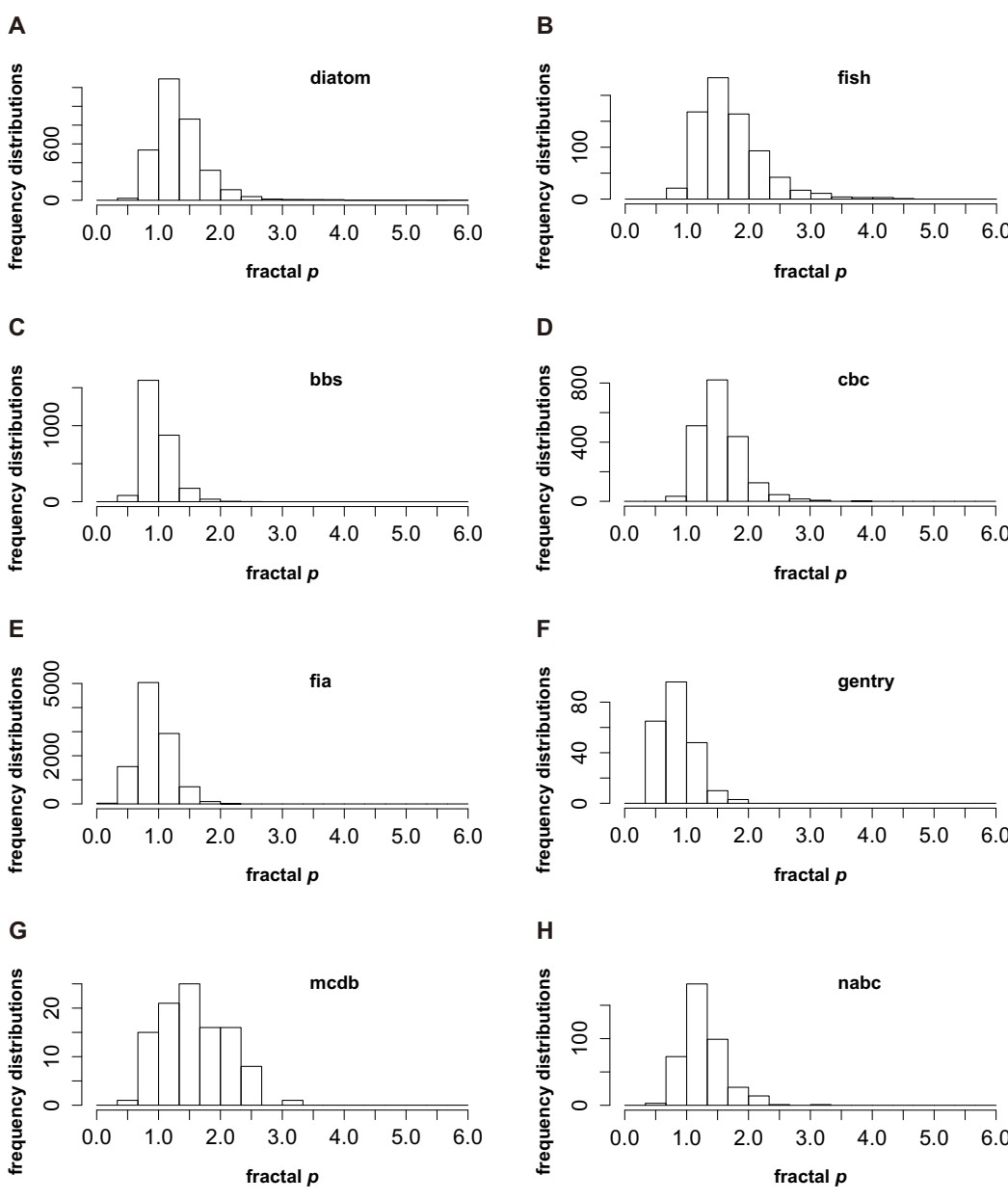

**Figure 1** The frequency distributions of the fractal $p$ (*Su, 2016*) for eight datasets (named "fish", "diatom", "nabc", "mcdb", "gentry", "fia", "cbc" and "bbs") from two sources (*Baldridge et al., 2016a*; *Baldridge et al., 2016b*; *Passy, 2016a*; *Passy, 2016b*). The peaks of $p$ frequencies for eight groups are not exactly the same. It could be from 0.67 to 1 (C, E and F), from 1 to 1.33 (A and H) and from 1.33 to 1.67 (B, D and G). However, the consensus of the $p$ frequencies for every group is very clear. The fractal $p$ mostly appears close to 1, and it is rare that $p$ is far greater than 1 or very near 0.

Secondly, it might be very hard to draw a definite conclusion about which pattern is the general SAD. On one hand, for the entire dataset, the average and median value of $p$ ($1.108 \pm 0.003$ and $1.034$) are both slightly higher than 1 (Table 1). The fractal $p$ of every group occurs frequently in a similar range (close to 1) (Fig. 1). On the other hand, the

peaks of $p$ distributions for eight groups are not exactly the same. It could be from 0.67 to 1 (Figs. 1C, 1E and 1F), or from 1.33 to 1.67 (Figs. 1B, 1D and 1G). Accordingly, this study assumes that the general pattern of SAD is that the fractal $p$ exceeds and approaches 1. This conjecture not only needs to be supported by further investigations and additional datasets, but also depends on how well it is explained in theory.

Finally, if the general pattern is that $p$ exceeds and approaches 1, how is it understood by current theories? In fact, such distribution ($p \approx 1$, see Eq. (1)) is approximately consistent with Zipf's law (or Zipf distribution) (*Zipf, 1949*; *Mandelbrot, 1953*; *Frontier, 1987*; *Seuront, 2009*). In communication systems, Zipf's law holds for almost all languages' letters and words (*Seuront, 2009*). Zipf stated that this empirical distribution attributed to the "Principle of Least Effort", representing a balance between the repetition desired by the listener and the diversity desired by the transmitter (*Zipf, 1949*; *Seuront, 2009*). If a repertoire is too repetitive, a communication is sent by a few signals, and less message is conveyed. Alternatively, if a repertoire is too diverse, the same message can be overrepresented by a multitude of signals, and less communication is conveyed. These opposite forces result in a balance between unification and diversification (*Zipf, 1949*; *Seuront, 2009*).

However, this principle seems hard to explain the general pattern of SAD from the perspective of ecology (*Mandelbrot, 1953*; *Frontier, 1987*). Later, Zipf's law was modified by Mandelbrot as $f_r = f_0(r + \beta)^{-\alpha}$ (*Mandelbrot, 1953*; *Frontier, 1987*; *Seuront, 2009*). In ecology, $f_r$ was the frequency of the $r$th species after ranking the species in decreasing order. $f_0$ is chosen such that the sum of all $f_r$, values predicted by the model is 1 (*Frontier, 1987*). The $\alpha$ and $\beta$ are conditioning the species diversity and the evenness of a given community (*Frontier, 1987*; *Seuront, 2009*). Unfortunately, the Mandelbrot model might be unable to fully explain the underlying mechanisms of the general SAD (*Mandelbrot, 1953*; *Frontier, 1987*; *Seuront, 2009*; *Su, 2016*).

This study proposes two hypotheses to elucidate the general pattern of SAD ($p$ exceeds and approaches 1). (1) The total abundance ($A_T$, see Eq. (4)) is equivalent to the total assimilated energy of the community, which is finite for a given condition. (2) The abundance of each species is linked with the energy transformation from the total energy, which increases the community entropy that determines the diversity. On one hand, the theoretical bases of two hypotheses actually derive from the first interpretation of the fractal model (see the second paragraph of 'Methods'), noting that a species is linked with the amount of assimilated energy (*Zipf, 1949*; *Mandelbrot, 1953*; *Frontier, 1987*; *Frontier, 1994*; *Mouillot et al., 2000*; *Su, 2016*). The noticeable differences between the first interpretation and two hypotheses are the finiteness of the total energy and the causality from the community entropy to the diversity. On the other hand, quantifying diversity according to the entropy (e.g., Shannon's entropy and Rényi's entropy) is not new (*Rényi, 1961*; *Hill, 1973*; *Magurran, 1988*; *Tuomisto, 2012*). Therefore, two hypotheses are not contradictory with current fractal and diversity theories (*Mandelbrot, 1953*; *Rényi, 1961*; *Hill, 1973*; *Frontier, 1987*; *Magurran, 1988*).

According to the first hypothesis, $A_T$ is finite. Thus, $A_T/A_1$ is also finite as $A_1$ is the abundance of dominant (see Eq. (1)). The finiteness of $A_T/A_1$ determines that the fractal

$p$ should be higher than 1 because $A_T/A_1$ converges for all $p > 1$ (see Eq. (4)). According to the second hypothesis, the diversity generally presents a trend of increasing because the energy conversion among species increases the community entropy. If the community entropy can be expressed as Hill's unifying notation and Rényi's entropy (*Rényi, 1961*; *Hill, 1973*; *Jost, 2006*; *Chao, Chiu & Jost, 2010*; *Jost, 2010*; *Chao, Chiu & Jost, 2014*; *Chao & Jost, 2015*), $A_T/A_1$ tends to increase with the entropy and diversity because $A_T/A_1$ is an effective number of species (*Rényi, 1961*; *Hill, 1973*). An increasing trend of $A_T/A_1$ means a decreasing trend of $p$ because the fractal $p$ is negative with $A_T/A_1$ (see Eq. (4)). Therefore, the balance between two forces (the fractal $p$ is higher than 1 and tends to decline) eventually leads to the general pattern of SADs that $p$ exceeds and approaches 1.

## CONCLUSIONS

When one investigates any SAD, previous studies suggested that it was ubiquitously observed that many species were rare and just a few were common (*May, 1975*; *Frontier, 1987*; *Magurran, 1988*; *Tokeshi, 1993*; *Mouillot et al., 2000*; *McGill et al., 2007*; *Baldridge et al., 2016a*; *Su, 2016*). However, this study indicates that it is more easily to observe that the fractal $p$ (*Su, 2016*) exceeds and approaches 1. This is the biggest difference between previous studies and this paper.

It might be a surprise that the fractal model has been around for some time but it is not widely used by ecologists (*Zipf, 1949*; *Mandelbrot, 1953*; *Frontier, 1987*; *Frontier, 1994*; *Mouillot et al., 2000*; *Su, 2016*). There were five families with over 40 SAD models, and it might be normal that some of them were not well known (*McGill et al., 2007*). *Tokeshi (1993)* commented that the fractal model was no more biological than others. However, such views were lack of empirical investigations (*Tokeshi, 1993*). In fact, there was a good fit of the fractal SAD to raw data (*Frontier, 1987*; *Frontier, 1994*; *Mouillot et al., 2000*; *Seuront, 2009*; *Su, 2016*), and its theoretical bases had been elaborated (*Frontier, 1985*; *Frontier, 1987*; *Frontier, 1994*; *Mouillot et al., 2000*). In the future, a more promising way is likely to evaluate the performance of the fractal model and increase the understanding of mechanisms that lead to the general pattern of SAD.

## ACKNOWLEDGEMENTS

My deepest gratitude goes to Marcio Pie (editor), Juan Pablo Gomez (reviewer) and an anonymous reviewer for their careful work and thoughtful suggestions that have helped improve this paper substantially.

### Funding

This work was supported by the National Natural Science Foundation of China (No. 41676113) and the Strategic Priority Research Program of the Chinese Academy of Sciences (Grant No. XDA13020102). The funders had no role in study design, data collection and analysis, decision to publish, or preparation of the manuscript.

# PeerJ

## Grant Disclosures

The following grant information was disclosed by the author:

National Natural Science Foundation of China: 41676113.

Strategic Priority Research Program of the Chinese Academy of Sciences: XDA13020102.

## Competing Interests

The authors declare there are no competing interests.

## Author Contributions

- Qiang Su conceived and designed the experiments, performed the experiments, analyzed the data, contributed reagents/materials/analysis tools, prepared figures and/or tables, authored or reviewed drafts of the paper, approved the final draft.

## Data Availability

The datasets named "fish" and "diatom" are from:

Passy SI (2016) Abundance inequality in freshwater communities has an ecological origin. The American Naturalist 187(4): 502–516.

https://doi.org/10.1086/685424.

The datasets named "nabc", "mcdb", "gentry", "fia", "cbc" and "bbs" are from:

Elita Baldridge, David J. Harris, Xiao Xiao, & Ethan White. (2016, November 15). weecology/sad-comparison: First revision for PeerJ (Version peerj.2). Zenodo. http://doi.org/10.5281/zenodo.166725.

## Supplemental Information

Supplemental information for this article can be found online at http://dx.doi.org/10.7717/peerj.5928#supplemental-information.

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
