# Peer review of "A general pattern of the species abundance distribution"

_PeerJ, doi:10.7717/peerj.5928_

## Round 0.1 · original submission · Major Revisions

I'd ask you to pay particular attention to the issue of goodness of fit of the de Zipf model identified by Reviewer 2.

Reviewer 1 ·

Basic reporting

Overall this ms is well written with clear language and well-organized structure. I also commend the author for including the data and the code in the submission. Below are a few minor issues that would require edits or additional clarification:
1. Line 27-30, "this study assumes ... when one investigates any SAD." I have trouble understanding what this sentence means. If Zipf's law for SAD is merely the author's assumption, why would one expect it in nature? Please revise.
2. Line 72, "there is (or is not)" - please change to "is there (or is there not)" for grammatical correctness.
3. Line 119, "Eq. (1) IS based on ...".
4. Line 198, "holds FOR".
5. Line 211, "f_0 is chosen such that the sum of f_r." This sentence is incomplete.
6. Lien 214, "understand" should be replaced with "explain" or a similar word (as the model itself can be understood but cannot understand).
7. Table 1. Please clarify what the range in the "Average" column (e.g., 0.008 in 1.343+-0.008) means. Is it standard deviation or standard error?
8. Reference, 2 and 3 are the same paper.
9. Line 222. What is "the first interpretation of the fractal model"? It is not clear from this paragraph.

Experimental design

The theoretical derivation is clearly laid out, and the data are adequately described. I would suggest the author to include an additional sentence or two at the end of the METHODS section to further explain how the data are analyzed to get the findings in results (i.e., how is p obtained), along the lines of "Ranked species abundances in each sample are fed into Eq. (2) to estimate the value of p."

Validity of the findings

While the analyses are valid, I think the interpretations can be further strengthened. It is perhaps not a surprise that the Zipf distribution fits SAD well (given the universally consistent shape of the SAD), and as the author acknowledged, the purpose of the ms is not to provide another comparison among SAD models. The empirical distribution of p (and the fact that it clusters around 1) is indeed more interesting. However, I do not fully agree with the author's reasoning in the last two paragraphs of the Discussion. The author's conclusion, that p exceeds and approaches 1, is partially based on the mathematical reasoning that the generalized harmonic series converges when p>1 and diverges when p<1 (Line 233-234). However, in reality r can never goes to infinity, since the number of species in a system is always finite. Thus even when p<1, the sum of r^(-p) (Eq 3) is still finite. In other words, there is no mathematical hard bound that forces p>1, which can be seen in Fig.1 where a significant proportion of the communities have p<1.

In addition, I do not fully follow the authors' argument for "a decreasing trend of p" (the second half of the last paragraph in Discussion). The author is correct to state that A_T/T_1 is positively correlated with diversity (Line 239), and that p is negatively correlated with A_T/A_1 (Line 242). However, p approaching 1 (the [incorrect] lower bound) would then imply that the ecological systems are somehow maximizing diversity. Please explain what is the theoretical basis for this implication.

A few additional minor changes that I would suggest the author to make to improve validity of the findings:
1. Line 89-91 provides two interpretations for the fractal model in ecology. I think this part is critically important to set the stage of the current student, and the two sentences included in the current ms are not sufficient. Please consider expanding this section into full paragraph(s) and elaborating on the ecological meaning of the two interpretations, to help readers better understand the context.
2. Line 127. The frequency distributions of SADs for these datasets are actually well known, since they have been extensively analyzed in previous studies, unless the author is referring to the frequency distribution of p when these datasets are fitted with the fractal model. Please rephrase.
3. An interesting finding that the author did not touch upon is the different distribution of p among studies/taxonomic groups (Fig.1). Can you explain/speculate whether the difference in distribution is due to differences in life history, behavior, sampling schemes, or something else?

Additional comments

Overall I think this ms is well written with valid analyses. However, the author's conclusion that the parameter p of the fractal model "exceeds and approaches 1" seems to be in contradiction to the results (Fig.1), where a large number of communities have p < 1. In addition, there are some inconsistencies in logic in the author's argument (the last paragraph in Discussion) to arrive at this conclusion. I would encourage the author to rethink and revise the discussion and strengthen the interpretations of the results.

·

Basic reporting

The study by Su, presents a very interesting problem in community ecology and in attempting to understand diversity patterns. As the author suggests, I agree that understanding the shape of the species abundance distribution is a crucial step to understanding the mechanisms regulating biodiversity and community assembly. A lot can be learned from studying SADs, but I also agree that it has been overall problematic. I enjoyed very much reading the manuscript as I found it very well written and easy to follow.

Although the manuscript is interesting and I applaud the intention of providing hypothesis for a general law in SADs, to me they are a bit confusing.I would like a little bit more explanation or maybe change the wording so that it the hypotheses are easily followed. I am not sure how the relationship between total abundance and energy assimilation is laid as a hypothesis for the shape of SADs in every community.

Experimental design

The research question is definitely relevant and the methods are well described and in sufficient detail. The manuscript is extremely clear and very reproducible.

Validity of the findings

I am concerned about the goodness of fit of the de Zipf model on each of the 19000 communities. There is no information in the manuscript regarding the goodness of fit and in this sense it is unclear to me if the distribution of the fractal parameter of the Zipf distribution is just a product of the generalized shape of the log-log relationship between the rank and the relative abundance of species. To clarify this point I performed a Monte Carlo simulation in which I generated 20000 communities, the first 10000 following a Zero-sum multinomial ddistribution with theta drawn at random from a uniform distribution between 10 and 50 and the size of the community drawn from a unifirom distribution between 10000 and 15000. I generated the second 10000 communities using a Poisson log-normal distribution with N=50, mean = 2 and sigma = 1. For each of these 20000 random communities I estimated the value o p. It turns out that the range of p resulted to be between 0.38 and 1.81 with mean 0.95 +- 0.22 and median 0.95. Even though the communities where generated with very different processes (one is product of a zero-sum game and the other one has been hypothesized to be the product of processes ranging from purely mathematical to niche based mechanisms). Does this mean that regardless of the generating process of the community assembly, the diversity and abundance of organisms is regulated by the amount of energy in the community and the transformation of the total energy by each species? Or is it totally and statistical artifact? To answer this, I think the solution would be to fit some alternative models and see if the Zipf with a fractal parameter of 1 is more likely or at least as likely as other generating models that have been previously proposed. If this turns out to be a real pattern then, I think the author has done an immense contribution to community ecology.

Additional comments

Minor comments.

line 211: Something seems to be missing here.
line 230: Should read "fractal and diversity theories"

---

## Round 0.2 · Minor Revisions

Both reviewers recognize your effort in accommodating their suggestions, but they still indicate a few issues that need to be addressed. Please pay particular attention to the issue raised by Reviewer 1 regarding the last paragraph of the discussion.

Reviewer 1 ·

Basic reporting

no comment

Experimental design

no comment

Validity of the findings

I respectfully disagree with the author's rebuttal regarding the last paragraph of the discussion, namely that p is constrained to be close to 1 because of two contrary forces at play.

In the first part of the argument, the author states that "p should be higher than 1", because "if sample size is not fixed in advance and can be allowed to be larger and larger, r might be infinity in theory". Mathematically, r going to infinity implies that species richness S is infinity. However this would never be the case. Richness is always a FINITE (albeit large and/or unknown) number even in theory, as long as our studies are confined on earth. And as long as S is finite, Eqn 3 always converges, regardless of whether p>1 or p<1. In other words, the statement that p should be higher than 1 is derived based on the incorrect assumption that richness can go to infinity.

In the second part of the argument, the author states that "the diversity generally presents a trend of (maximizing diversity) because the energy conversion among species increases the community entropy." I still find this statement quite confusing, and I think would benefit future readers of the paper if the author could elaborate on why this is the case. Does the author mean that the ecological system is maximizing entropy? Or entropy production? Is there a reason why the ecological system is expected to have this kind of behavior?

·

Basic reporting

As I mentioned in the first review, the paper is in general well written and very clear. It is overall easy to understand and provides enough background of the problem. I would suggest to check a couple of places where there are some grammatical mistakes. For example,
Line 109, is estimated as follows
Line 113, I would delete line 113. I understand that this is in response to reviewer 1 but to me is chela enough in line 108 that equation 2 is used to obtain the MLE of p for each SAD.
Line 261, replace most by strongest or biggest.

Experimental design

No Comment

Validity of the findings

I feel that I would like to see more discussion about the reason why all communities are expected to have a fractal parameter of 1 irrespective of the community assembly process. The author has a very interesting paragraph explaining this that I would like to see added to the discussion. I also encourage the author to add a statement about the R2. I apologized that I missed it the first time but there was no reference in the main text that R2 was calculated for each SAD as a goodness of fit measurement. In my opinion the table that the author added as supplementary material and shows in the rebuttal letter should be also added to the main text.

Additional comments

First of all thank you very much for the honoring offer of working together. I would love to collaborate with the author in future studies attempting to understand how do different mechanisms of community assembly can result in similar SAD predicted by the same fractal parameter. Also I would love to explore what is the relation of the energy transfer and the community assembly process.
Second, I am glad to see that the author does not think that the findings are an statistical mistake. To me it makes now sense that the overall energy in a community determines the diversity irrespective of the community assembly process. However, there seems to be something that I am still not understanding. The author shows in one of his previous papers (Su et al 2016.) that the fractal parameter is related to other diversity indices. In general, you can think about this diversity indices to be a measurement of species richness and the distribution of abundance across species, (i.e. how homogeneous or heterogeneous is the distribution of abundance across species.). This means that in general, the larger the diversity index, the higher the species richness (this is not necessarily true when the abundance is homogeneously distributed across species), which necessarily translates to the fact that the higher the fractal parameter the higher the species richness of a community. How can the author reconcile the huge differences in species richness in communities across the globe with the fact that the fractal parameter in every community should be around 1, or close to 1? Is there something in this explanation that I am missing? I think that the author needs to expand a little the discussion keeping in mind the natural history and the implications that this findings have into understanding the regulation of biodiversity. What happens for example in SAD of communities that have been recently invaded by organisms which have a very high abundance compared to their native places? How can you explain that a community in the tropics with hundreds of species will follow the same rules of a community in the temperate region with just tens of species? Obviously, the energy available for the community is much lower in the temperate region that in tropical regions but then, how does a community trees in a boreal forest can have the same fractal parameter than a tree community in the amazon with at least 1000 species? I am not sure how the increasing trend of community entropy is not related to the generating processes when the generating processes such as speciation, drift and selection necessarily change the entropy of the system by introducing new species, selecting for better adapted ones and randomly monopolizing space? The author states that "the entropy of the community increases with the process of energy transfer" but would this not mean that this results in a larger fractal parameter? If shannon and simpson indices are entropy indices and the fractal parameter is related to them, why would you expect that the fractal parameter is the same for any community in the world? I do think that the findings are very interesting but given the sensitivity of the topic and the importance of the results i would encourage the author to explain this issues in more detail in the discussion so that people more familiar with community assembly mechanisms can understand the interpretation of these results better.

---

## Round 0.3 · accepted · Accept

I think you properly addressed all of the remaining issues. Congratulations!

#